# Improving Seq2Seq Grammatical Error Correction
# via Decoding Interventions

**Houquan Zhou🍊, Yumeng Liu🍊, Zhenghua Li🍊,✉ Min Zhang🍊**
**Bo Zhang🍵, Chen Li🍵, Ji Zhang🍵, Fei Huang🍵**
🍊 Institute of Artificial Intelligence, School of Computer Science and Technology,
Soochow University, China;
{hqzhou,ymliu14}@stu.suda.edu.cn, {zhli13,minzhang}@suda.edu.cn
🍵 DAMO Academy, Alibaba Group, China
{klayzhang.zb,puji.lc,zj122146,f.huang}@alibaba-inc.com

## Abstract

The sequence-to-sequence (Seq2Seq) approach has recently been widely used in grammatical error correction (GEC) and shows promising performance. However, the Seq2Seq GEC approach still suffers from two issues. First, a Seq2Seq GEC model can only be trained on parallel data, which, in GEC task, is often noisy and limited in quantity. Second, the decoder of a Seq2Seq GEC model lacks an explicit awareness of the correctness of the token being generated. In this paper, we propose a unified decoding intervention framework that employs an external critic to assess the appropriateness of the token to be generated incrementally, and then dynamically influence the choice of the next token. We discover and investigate two types of critics: a pre-trained left-to-right language model critic and an incremental target-side grammatical error detector critic. Through extensive experiments on English and Chinese datasets, our framework consistently outperforms strong baselines and achieves results competitive with state-of-the-art methods.

## 1 Introduction

Automatically correcting grammatical errors is an important task of practical value in the NLP field. The potential applications include document proof-reading, writing assistant, language learning education, text post-processing for automatic speech recognition (Leng et al., 2021), etc. There are two mainstream approaches to grammatical error correction (GEC), namely sequence-to-sequence (Seq2Seq) (Sun et al., 2021; Rothe et al., 2021) and sequence-to-edit (Seq2Edit) (Awasthi et al., 2019; Omelianchuk et al., 2020). The Seq2Seq approach treats GEC as a monolingual text translation/transduction task, whereas the Seq2Edit approach casts GEC into a sequence labeling task.

---

✉: Zhenghua Li is the corresponding author.

**Input:** But there *had* no buyers .
**Reference:** But there were no buyers .

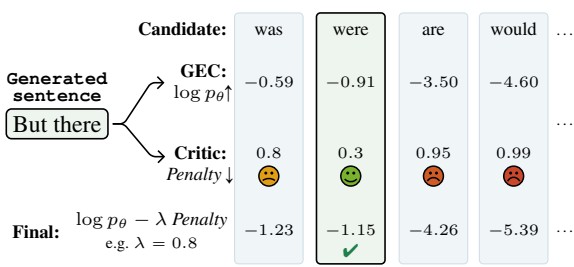

Figure 1: Decoding intervention uses a critic to score the correctness of the token attaching to the partially generated target sentence. The final score of a candidate token is the log-probability from the GEC model subtracted by the critic penalty, which is scaled by $\lambda$.

Recent studies show that the Seq2Seq approach consistently outperforms the Seq2Edit approach on a variety of languages and datasets, especially in handling more complex errors such as word-ordering ones (Qorib et al., 2022; Zhang et al., 2022b). However, the Seq2Seq approach still suffers from two issues.

First, a Seq2Seq model can only utilize parallel sentence pairs as training data, in which the input sentence is potentially ungrammatical whereas the target one is considered as correct. Usually, a major proportion of the training data is automatically collected from language learner websites such as Lang-8. On the one hand, the Lang-8 data contains a certain amount of noises, considering the voluntary contributors may make mistakes as well. On the other hand, the data scale is quite limited, compared with the non-parallel data used for training large language models. For instance, the cleaned version of English Lang-8 corpus (CLang8) contains only 2.4M sentence pairs (Rothe et al., 2021).

Data augmentation is a popular approach for addressing the limited scale issue, i.e., synthesizing

large amount of training data (Grundkiewicz et al., 2019; Stahlberg and Kumar, 2021). However, it can be very difficult and tricky to control the error distribution in the generated data so that it resembles the realistic scenario. Moreover, it brings a heavy computational cost to train a GEC model with very large training data.

The second issue of the Seq2Seq GEC model is that the decoder lacks an explicit awareness or evaluation of whether the generated token is correct during decoding. There are indeed several works that perform grammatical error detection (GED) for the input sentence, and use the results as extra features for the encoder so that the decoder pays extra attention to the erroneous spans in an implicit manner (Chen et al., 2020; Yuan et al., 2021; Zhang et al., 2022b). However, we are not aware of any previous works that explicitly checks the correctness of generated tokens during decoding (e.g., target-side GED). As pointed out by Mita and Yanaka (2021), a Seq2Seq GEC model tends to generate wrong corrections when the model encounters errors unseen in the training data.

In this work, we propose a decoding intervention framework to address both issues of the Seq2Seq GEC approach. As illustrated in Figure 1, we employ an external **critic** to assess the appropriateness of the token to be generated **incrementally**, and then **dynamically** influence the choice of the next token. Specifically, at each decoding step, the critic will evaluate the appropriateness of the candidate tokens, if the token is inappropriate, the critic will punish the generation of the token by reducing the log-probability score of the token.

The key to our decoding interventions is to find a suitable critic. We discover and investigate two useful critics. The first critic is a pre-trained left-to-right language model (LM). Using the language model as the critic can take advantage of its knowledge learned from the vast amount of text. If the language model gives a low probability to a token, then the token is probably wrong even if the GEC model gives it a high probability. The second critic is a GED model which is an ideal critic to incorporate the explicit awareness of correctness into the Seq2Seq GEC model during the decoding process. However, the conventional GED cannot be directly used as the critic because it does not match the incremental manner of the decoding process. To address this problem, we propose an incremental target-side GED, which acts in a Seq2Seq manner

making judgments on the token to be generated $y_t$ based on both the input sentence $\boldsymbol{x}$ and the tokens generated so far $\boldsymbol{y}_{<t}$.

We conduct experiments on three English GEC datasets, including two English-as-a-second-language (ESL) datasets, a multi-domain native-English dataset, and a Chinese dataset. Experimental results demonstrate that our decoding intervention brings consistent and substantial improvements on all datasets. The results also show that with the help of decoding intervention, our GEC model can achieve comparable performance to the state-of-the-art models on all datasets under comparable settings without any re-training.

Our code is available at https://github.com/Jacob-Zhou/gecdi

## 2 The Basic Seq2Seq GEC Model

This work aims to improve the Seq2Seq GEC approach. In this section, we briefly describe it. Given a potentially erroneous sentence, a Seq2Seq GEC model tries to generate a correct one without changing its meaning, similar to machine translation, yet in a monolingual fashion.

We adopt the widely-used Transformer architecture (Vaswani et al., 2017) as our model backbone, which comprises an encoder and a decoder. Given an input sentence $\boldsymbol{x} = x_1 \ldots x_n$, the encoder first encodes it into a sequence of hidden states $\boldsymbol{h} = h_1 \ldots h_n$. At each timestamp, given the input sentence representation $\boldsymbol{h}$ and the previously generated tokens $\boldsymbol{y}_{<t}$, the decoder calculates a probability distribution over the vocabulary for the next-token generation:

$$p_\theta(y_t \mid \boldsymbol{y}_{<t}, \boldsymbol{x}) = \text{Decoder}(\boldsymbol{y}_{<t}, \boldsymbol{h}). \quad (1)$$

The score of an output sentence $\boldsymbol{y}$ is the sum of the log-probabilities of all predicted tokens:

$$s(\boldsymbol{x}, \boldsymbol{y}) = \sum_{t=1}^{|\boldsymbol{y}|} \log p_\theta(y_t \mid \boldsymbol{y}_{<t}, \boldsymbol{x}). \quad (2)$$

During training, Seq2Seq models commonly employ the teacher forcing method, aiming to maximize the log-likelihood of the ground-truth next token $g_t$, given the input sentence $\boldsymbol{x}$ and the previous ground-truth tokens $\boldsymbol{g}_{<t}$:

$$\mathcal{L}(\theta, \boldsymbol{x}, \boldsymbol{g}) = -\sum_{t=1}^{|\boldsymbol{g}|} \log p_\theta(g_t \mid \boldsymbol{g}_{<t}, \boldsymbol{x}). \quad (3)$$

The main advantage of teacher forcing is that it allows for parallel training.

The inference of Seq2Seq GEC models is to find the best output sentence $\boldsymbol{y}^*$ by solving the following optimization problem:

$$\boldsymbol{y}^* = \arg\max_{\boldsymbol{y} \in \mathcal{Y}} s(\boldsymbol{x}, \boldsymbol{y}), \qquad (4)$$

where $\mathcal{Y}$ is the set of all possible sentences. This optimization problem is typically tackled using the beam search algorithm, in which the model predicts a token at each decoding step, appends it to the partial sentence, and subsequently selects the top $k$ partial sentences based on their scores for the next decoding step.

## 3 Decoding Intervention

In this work, we propose a decoding intervention framework to improve the Seq2Seq GEC. Concretely, we use an external critic model to dynamically evaluate the correctness of the next token predicted by the GEC model during decoding. The evaluation produces a penalty score to the existing probability distribution from the GEC model as follows:

$$s(\boldsymbol{x}, \boldsymbol{y}) = \sum_{t=1}^{|\boldsymbol{y}|} \big( \log p_\theta(y_t \mid \boldsymbol{y}_{<t}, \boldsymbol{x}) \\ - \lambda \times Penalty(y_t, \boldsymbol{y}_{<t}, \boldsymbol{x}) \big). \qquad (5)$$

The first term is the original probability from the GEC model. The logarithm transform stretches the probability into a wider range and thus makes it more influential. This also gives more flexibility to the design of the critic model[1].

The second term is the penalty score from the critic model to "$y_t$" given the input sentence $\boldsymbol{x}$ and the generated prefix $\boldsymbol{y}_{<t}$ and $\lambda$ is a coefficient that controls the trade-off between two model scores. Please note that $\lambda$ is not a global hyper-parameter but is instead decided by the scores in a token-wise manner. We detail this in Section 3.3.

From Eq. (5), we can draw two characteristics of our framework.
- *Incremental*. Similar to the Seq2Seq GEC model, the critic model incrementally evaluates a target sentence from left to right, token by token.

---

[1]Based on our early-stage trials, we find it problematic to directly integrate the probabilities of the GEC model and the critic model via weighted interpolation, since the models usually have different vocabulary spaces and smaller vocabulary leads to a relatively larger probability for each token.

- *Dynamic*. The critic model dynamically influences the choice of tokens during decoding, in contrast to re-ranking $N$ complete sentences.

Moreover, the critic model may or may not use the input sentence $\boldsymbol{x}$. In this work, we discover and investigate two useful critic models, i.e., a pure left-to-right pre-trained language model which does not use $\boldsymbol{x}$, and an incremental target-side GED model that uses $\boldsymbol{x}$.

### 3.1 Left-to-Right Pre-trained LM

A conventional pre-trained left-to-right language model, unlike masked language models (e.g., BERT (Devlin et al., 2019)) and Seq2Seq models (e.g., BART (Lewis et al., 2020)), can be naturally used to evaluate the possibility of a sentence $\boldsymbol{y}$, which is factored as product of probabilities of tokens in an incremental manner.

$$p_\pi(\boldsymbol{y}) = \prod_{t=1}^{|\boldsymbol{y}|} p_\pi(y_t \mid \boldsymbol{y}_{<t}), \qquad (6)$$

where $\pi$ denotes parameters of the language model. The possibility of a token, i.e., $p_\pi(y_t \mid \boldsymbol{y}_{<t})$, can also be understood as how likely the token appears after previous tokens $\boldsymbol{y}_{<t}$.

The GEC task aims to produce a correct sentence that keeps the same meaning as the input sentence. We propose to use a language model to evaluate the correctness of a sentence from a purely linguistic perspective, without referring to its input sentence.

In this work, we select the GPT-2 models, which are trained on a very large amount of sentences, much more than the parallel sentences that are used for training GEC models, as the pure left-to-right language models. The rationale is that if the language model gives a low probability to a token, then the token is probably wrong even if the GEC model gives it a high probability. Specifically, we define the penalty from the language model critic as follows.

$$Penalty^{lm}(y_t, \boldsymbol{y}_{<t}, \boldsymbol{x}) = 1 - p_\pi(y_t \mid \boldsymbol{y}_{<t}) \qquad (7)$$

### 3.2 Incremental Target-side GED

As discussed in Section 1, one potential weakness of Seq2Seq GEC models is that the decoder may be unaware of the correctness of its output tokens. Several recent works try to alleviate this issue by performing GED on the input sentence, and using the GED labels as extra inputs to the encoder (Chen et al., 2020; Yuan et al., 2021; Zhang et al., 2022b).

| Type | Generated Tokens | Next |
|------|------------------|------|
| COR  | The quick brown fox | **jumps** |
| RED  | The quick brown fox | *already* |
| SUB  | The quick brown fox | *runs* |
| MISS | The quick brown fox | *over* |

Table 1: Examples of the four GED labels, i.e., correct (COR), redundant errors (RED), substitution errors (SUB), and missing errors (MISS). In this example, the input sentence is "The quick brown fox *jump* over the lazy dog", and the reference is "The quick brown fox **jumps** over the lazy dog".

To some extent, this approach can make the model more explicitly aware of the correction process. In this work, we for the first time propose to apply an incremental target-side GED to the output sentence under our framework, which we believe is a more effective intervention strategy.

Given an input sentence $x$, a partial target sentence generated so far $y_{<t}$, and a candidate token $y_t$ to be generated, the GED model judges the correction of $y_t$ into four labels, as shown in Table 1. Please notice that the GED model must look at $x$, instead of only accessing $y_{<t}$. The reason is that the GED model as a critic provides a complementary impact versus the language model critic that the target sentence should keep the same meaning as the input sentence. In the absence of $x$, many tokens can be considered correct given $y_{<t}$.

Formally, we design the penalty from the GED critic model as follows.

$$Penalty^{ged}(y_t, \boldsymbol{y}_{<t}, \boldsymbol{x}) = 1 - p_\Phi(\text{COR} \mid \boldsymbol{y}_{\leqslant t}, \boldsymbol{x}),$$
(8)

where $\Phi$ is the model parameters, and "COR" is the correct label as shown in Table 1.

**Training**  Our incremental target-side GED acts in a Seq2Seq manner, which is much like the GEC model, requiring parallel sentence pairs for training. Yet we cannot directly use the GEC training data, because the target sentences are all correct. The other consideration is that it is obviously beneficial that errors in the target sentences are more consistent with those generated by GEC models. Basically, we use the baseline GEC model to generate $K$ output sentences which may be erroneous via beam search. Then we obtain the error labels for each token (subword, to be accurate) using the editing distance algorithm, which is the same as the evaluation metrics. Section 4 gives more details.

### 3.3 Coefficient for the Critics

The coefficient $\lambda$ in Eq. (5) is important for leveraging the power of the critic model. Instead of using a fixed value for all contexts, we find it is beneficial to dynamically set the value by comparing the confidences from the two participating models. Intuitively, a model can be trusted if it has high confidence in its prediction, as a strong correlation holds between a model's confidence and the accuracy of its prediction (Guo et al., 2017; Kull et al., 2019). After several experimental trials, we find the following formula works quite well, especially when we use two critics at the same time. Here we use the language model critic for illustration.

$$\lambda = \alpha \times \frac{\beta \times Entropy(\ p_\theta(\cdot)\ ) + 1}{\beta \times Entropy(\ p_\pi(\cdot)\ ) + 1}.$$
(9)

where $p_\theta(\cdot)$ and $p_\pi(\cdot)$ refer to the probability distribution of the GEC model and the language model over their own vocabulary space $\mathcal{V}$; $\alpha > 0$ is a coefficient that controls the overall scale of penalty scores, and $\beta \geqslant 0$ governs their variation, both of which aim to balance the influence of the critic models.

For the GED critic, we simply replace $p_\pi(\cdot)$ with $p_\Phi(\cdot)$. Please notice that its vocabulary only contains the four GED tags[2].

We separately select $\alpha$ and $\beta$ for the two critics based on dev data. The search space of $\alpha$ is $\{0.1, 0.2, \ldots, 1.0\}$ and that of $\beta$ is $\{0.01, 0.1, \ldots, 100\}$. In Section 5, we study the impact of $\alpha$ and $\beta$. Results show that our decoding intervention is robust on a wide range of $\alpha$ and $\beta$.

**Using both critics**  When we use two critics at the same time, we directly add penalties from the two critic models in Eq. (5).

$$\ldots - \lambda^{lm} \times Penalty^{lm}(y_t, \boldsymbol{y}_{<t}, \boldsymbol{x})$$
$$- \lambda^{ged} \times Penalty^{ged}(y_t, \boldsymbol{y}_{<t}, \boldsymbol{x}))$$
(10)

For simplicity, we directly re-use the hyperparameters separately selected above, and find the performance is satisfactory.

## 4   Experiments

**Datasets**  In this paper, we conduct experiments on two languages: *English* and *Chinese*.

---

[2]To avoid the entropy imbalance resulting from different vocabulary sizes between the two critics, we normalize the entropies to a range of $[0, 1]$. This normalization is achieved by dividing by the upper bound of the entropy ($\log|\mathcal{V}|$).

| System | English | | | | | | | | | Chinese | | |
|---|---|---|---|---|---|---|---|---|---|---|---|---|
| | CoNLL-14 *test* | | | BEA-19 *test* | | | GMEG_WIKI *test* | | | MuCGEC *test* | | |
| | P | R | $F_{0.5}$ | P | R | $F_{0.5}$ | P | R | $F_{0.5}$ | P | R | $F_{0.5}$ |
| Omelianchuk et al. (2020) | 77.5 | 40.1 | 65.3 | 79.2 | 53.9 | 72.4 | – | – | – | – | – | – |
| Rothe et al. (2021) | – | – | 66.1 | – | – | 72.1 | – | – | – | – | – | – |
| Sun et al. (2021) | 71.0 | 52.8 | 66.4 | – | – | 72.9 | – | – | – | – | – | – |
| Yasunaga et al. (2021) | 78.0 | 40.6 | 65.8 | 79.4 | 55.0 | 72.9 | 57.9 | 33.6 | 50.6[†] | – | – | – |
| Sun and Wang (2022) | – | – | – | 78.7 | 63.2 | **75.0** | – | – | – | – | – | – |
| Zhang et al. (2022b) | 74.7 | 49.0 | **67.6** | 75.1 | 65.5 | 72.9 | – | – | – | 54.69 | 29.10 | **46.51** |
| Vanilla Decoding | 76.1 | 48.3 | 68.2 | 76.3 | 60.7 | 72.5 | 72.3 | 34.7 | 59.4 | 56.13 | 29.38 | 47.41 |
| Decoding Intervention | | | | | | | | | | | | |
| ⊢ Language Model | 77.0 | 48.6 | 68.9 | 76.2 | 61.1 | 72.6 | 72.7 | 35.5 | 60.1 | 55.55 | 30.51 | 47.66 |
| ⊢ Target-side GED | 78.6 | 46.3 | 69.0 | 77.5 | 59.5 | 73.0 | 75.0 | 33.5 | 60.1 | 56.94 | 30.06 | 48.24 |
| ⊢ Both | 79.2 | 46.8 | **69.6** | 77.4 | 59.9 | **73.1** | 75.6 | 34.4 | **61.0** | 56.74 | 31.00 | **48.61** |

Table 2: Results on GEC test datasets. †: The model of Yasunaga et al. (2021) in GMEG_WIKI dataset is only trained on synthetic data, which makes direct comparisons less meaningful.

For English, we follow the convention of using BEA-19 dev set (Bryant et al., 2019) for method-ological development and the BEA-19 and CoNLL-14 test sets for final evaluation. It should be noted that both the BEA-19 and CoNLL-14 test sets are collected from ESL learners. To better understand the effectiveness of our method in real-world scenarios, we also conduct experiments on GMEG-wiki dev/test set (Napoles et al., 2019), a multi-domain dataset derived from native English speakers. For performance metrics, we use $M^2$Scorer on CoNLL-14 and ERRANT v2.0.0 on the others.

For Chinese, we conduct experiments on the MuCGEC dataset (Zhang et al., 2022a), a multi-reference and multi-source dataset[3] and use the official ChERRANT scorer to measure the performance.

**Baseline & general settings** The GEC model used in this paper is a BART model (Lewis et al., 2020) fine-tuned on GEC datasets. Detailed information on this model can be found in Appendix B.

We take "Vanilla Decoding" as our baseline, which refers to decoding using the original probability score as defined in Eq. (2).

During the decoding process, we employ the commonly used beam-search algorithm to find the sequence with the highest score $s(\boldsymbol{x}, \boldsymbol{y})$. For all experiments, we use a beam size of 12.

We repeat all the experiments 4 times with different random seeds and report the average results.

**Language model critic** We take off-the-shelf GPT-2 models as our language model critics. For

ESL datasets, we use the `gpt2` model, while for the GMEG-wiki dataset, we opt for the larger `gpt2-large` model. For the Chinese dataset, MuCGEC, we employ the `uer/gpt2-chinese-cluecorpussmall`.

**Target-side GED critic** We initialize the backbone of our target-side GED critic models with pre-trained BART models. Specifically, we use the `facebook/bart-base` for ESL datasets, the larger `facebook/bart-large` for the GMEG-wiki dataset, and the `fnlp/bart-large-chinese` for the MuCGEC dataset.

We use the FCE, NUCLE, W&I+LOCNESS train set to generate the English training data. And, we use the HSK train set (Zhang, 2009) for Chinese critic models training[4]. Hyper-parameter details can be found in Appendix C.

### 4.1 Main Results

The main results are presented in Table 2. Results show that compared to the baseline "Vanilla Decoding", our decoding intervention consistently improves $F_{0.5}$ scores across all datasets, regardless of the critic used. The two critics improve the model's performance in different ways. The language model critic is better at improving the recall rate, while the target-side GED critic is better at improving the precision rate. Results also show that our decoding intervention can be further improved by combining the two critics ("**Both**"). Specifically, "Both" achieves 1.4, 0.6, 1.6, and 1.2 $F_{0.5}$ improvement on the CoNLL-2014, BEA-19,

---

[3]Please note that we omit the experiments on the NLPCC-18 (Zhao et al., 2018) since it is included in MuCGEC.

[4]We allocate 90% of the generated data for training and 10% for development.

| Input | Scientists can not conclude whether this smell tactic is used to attract the Danman (evil partner in crime whom has mad guitar skillz) or to ward of predators (PhD supervisors) . |
|---|---|
| Reference 0 | . . . crime **who** has mad guitar skillz) or to ward of predators (PhD supervisors) . |
| Reference 1 | . . . crime **who** has mad guitar **skills**) or to ward **off** predators (PhD supervisors) . |
| Vanilla Decoding | . . . crime **who** has mad guitar **skills**) or to ward *of* predators (PhD supervisors) . |
| Decoding Intervention | |
| ├ Language Model | . . . crime **who** has mad guitar **skills**) or to ward **off** predators (PhD supervisors) . |
| ├ Target-side GED | . . . crime **who** has mad guitar **skills**) or to ward *of* predators (PhD supervisors) . |
| └ Both | . . . crime **who** has mad guitar **skills**) or to ward **off** predators (PhD supervisors) . |
| Input | Girls were first admitted to Hurlstone Agricultural High School in 1978 . |
| Reference | Girls were first admitted to Hurlstone Agricultural High School in 1978 . |
| Vanilla Decoding | ***The*** girls were first admitted to Hurlstone Agricultural High School in 1978 . |
| Decoding Intervention | |
| ├ Language Model | ***The*** girls were first admitted to Hurlstone Agricultural High School in 1978 . |
| ├ Target-side GED | Girls were first admitted to Hurlstone Agricultural High School in 1978 . |
| └ Both | Girls were first admitted to Hurlstone Agricultural High School in 1978 . |

Table 3: Qualitative examples of decoding intervention versus vanilla decoding. Corrections marked in "**Blue**" are correct or suggested by the reference, while those in "*Red*" are incorrect.

GMEG-wiki, and MuCGEC test sets, respectively.

We also compare our model with the recent state-of-the-art models. Note that our baseline model is already competitive with the state-of-the-art models. The tricks that we have used to improve the baseline model's performance are listed in Appendix B.3. Results show that our decoding intervention method ("Both") achieves an absolute improvement of 2.0 $F_{0.5}$ on the CoNLL-2014 and 2.10 on the MuCGEC. It is worth noting that the best performance in the BEA-19 test is achieved by Sun and Wang (2022) with an $F_{0.5}$ score of 75.0. However, it can not be directly compared with our results since they use a private synthesized dataset and the size of it is hundreds of times larger than our training data (300M vs. our 2.4M).

## 4.2 Qualitative Examples

We include two qualitative examples in Table 3.

In the first example, the baseline "Vanilla Decoding" and the decoding intervention using the target-side GED as the critic both fail to correct the error "*ward of*" to "*ward off*". It is because the error pattern ("*ward of*" to "*ward off*") has not appeared in the training data of both the GEC model and the target-side GED. However, the language model critic is able to correct this error successfully, demonstrating that a language model, pre-trained on vast amounts of data, can help the GEC model identify and correct errors that do not appear in the GEC training data.

In the second example, the input sentence is grammatically correct. Yet, the baseline "Vanilla Decoding" introduces a new error by inserting a definite article "*The*" before "*Girls*". The language

model critic fails to correct this by intervening in the decoding process, since the sentence with the definite article is still grammatically correct, albeit with a different meaning.

These two examples also show that the "**Both**", which uses the target-side GED and the language model at the same time, manages to integrate the advantages of both critics.

## 5 Ablation Studies

**Impact of critic sizes** We perform experiments using four distinct sizes of language models and two different sizes of target-side GEDs.

As shown in Table 4, we can observe that, on BEA-19, the ESL learners dataset, a larger critic only results in a slight improvement in the $F_{0.5}$ score (+0.07 for language models, and +0.08 for target-side GEDs). However, on the GMEG-wiki, a multi-domain dataset from native speakers, a larger critic can lead to a large improvement on $F_{0.5}$ score (+0.30 for language models, and +0.76 for target-side GEDs). This may be because the errors on the ESL dataset are relatively simple and can be captured by smaller critics. In contrast, errors on the multi-domain native dataset are more complex and may require domain knowledge to identify.

Due to the uniform size of the Chinese GPT-2 models we found, we only performed experiments on the target-side GEDs for the Chinese dataset. The results show that a larger target-side GED is more effective when used with the Chinese dataset.

**Effectiveness of the dynamic coefficient** As mentioned in Section 4.1, the language model and the target-side GED exhibit specific tendencies

| | Size | P | R | $F_{0.5}$ |
|---|---|---|---|---|
| **BEA-19 *dev*** | | | | |
| Vanilla Decoding | | 64.85 | 39.44 | 57.40 |
| Decoding Intervention | | | | |
| ├ Language Model | | | | |
| │ ├ gpt2 | 117M | 65.12 | 39.83 | 57.74 |
| │ ├ gpt2-medium | 345M | 65.10 | 39.65 | 57.64 |
| │ ├ gpt2-large | 774M | 65.10 | **39.92** | 57.76 |
| │ └ gpt2-xl | 1.5B | **65.30** | 39.75 | **57.81** |
| └ Target-side GED | | | | |
| ├ bart-base | 110M | **66.52** | 37.86 | 57.72 |
| └ bart-large | 400M | 66.01 | **38.71** | 57.80 |
| **GMEG**WIKI ***dev*** | | | | |
| Vanilla Decoding | | 70.21 | 32.85 | 57.19 |
| Decoding Intervention | | | | |
| ├ Language Model | | | | |
| │ ├ gpt2 | 117M | 70.76 | 33.31 | 57.76 |
| │ ├ gpt2-medium | 345M | 70.91 | 33.36 | 57.88 |
| │ ├ gpt2-large | 774M | **71.13** | 33.50 | **58.07** |
| │ └ gpt2-xl | 1.5B | 71.07 | **33.54** | 58.06 |
| └ Target-side GED | | | | |
| ├ bart-base | 110M | 71.16 | **32.10** | 57.22 |
| └ bart-large | 400M | **73.01** | 31.84 | **57.98** |
| **MuCGEC *dev*** | | | | |
| Vanilla Decoding | | 55.69 | 28.87 | 46.88 |
| Decoding Intervention | | | | |
| └ Target-side GED | | | | |
| ├ bart-base | 139M | **56.44** | 28.60 | 47.15 |
| └ bart-large | 406M | 55.82 | **29.83** | 47.48 |

Table 4: Results on GEC dev datasets of different sizes of GPT-2 and target-side GED models.

| | P | R | $F_{0.5}$ |
|---|---|---|---|
| **BEA-19 *dev*** | | | |
| Vanilla Decoding | 64.85 | 39.44 | 57.40 |
| Decoding Intervention | | | |
| ├ Language Model | 65.30 | 39.75 | **57.81** |
| │ └ w/o dynamic coefficient | 64.33 | **40.30** | 57.43 |
| └ Target-side GED | 66.01 | 38.71 | 57.80 |
| └ w/o dynamic coefficient | 66.09 | 38.59 | 57.79 |
| **GMEG**WIKI ***dev*** | | | |
| Vanilla Decoding | 70.21 | 32.85 | 57.19 |
| Decoding Intervention | | | |
| ├ Language Model | 71.13 | **33.50** | **58.07** |
| │ └ w/o dynamic coefficient | 70.23 | 33.08 | 57.34 |
| └ Target-side GED | 73.01 | 31.84 | 57.98 |
| └ w/o dynamic coefficient | **73.44** | 31.49 | 57.95 |
| **MuCGEC *dev*** | | | |
| Vanilla Decoding | 55.69 | 28.87 | 46.88 |
| Decoding Intervention | | | |
| ├ Language Model | 55.88 | **30.02** | **47.59** |
| │ └ w/o dynamic coefficient | 55.85 | 30.00 | 47.57 |
| └ Target-side GED | 55.82 | 29.83 | 47.48 |
| └ w/o dynamic coefficient | **56.43** | 28.68 | 47.20 |

Table 5: Results on GEC dev datasets of w/ or w/o dynamic coefficient strategy. Underline means the result is inferior to the vanilla decoding baseline.

when improving the GEC model. However, we also observed that a critic tends to decrease one score when improving another. For instance, while the target-side GEDs improve the precision score, they also result in a decline in the recall score. It might be caused by the misjudgment of the critics when they are unconfident. As a result, the improvement of the $F_{0.5}$ score is potentially hindered. To address this issue, we propose a coefficient strategy that dynamically adjusts the coefficient of the critic at each decoding time step according to the confidence levels of the critic and the GEC model.

Results in Table 5 show that the dynamic coefficient strategy can alleviate the decrease in either the precision score or recall score. Furthermore, this strategy can even lead to an improvement of the P score on the BEA-19 dataset when using the language model as the critic and an improvement of the R score on the MuCGEC dataset when employing the target-side GED as the critic.

**Robustness of the decoding intervention** The dynamic coefficient strategy of the decoding in-

tervention contains two hyper-parameters: an $\alpha$ for controlling the global scale of the coefficient and a $\beta$ for the coefficient's variability. We use a heatmap to visualize the impact of these two hyper-parameters on the $F_{0.5}$ score, as shown in Figure 2.

In general, our decoding intervention is robust to the hyper-parameters, and surpasses the baseline in most cases. However, we can also observe some interesting phenomena. Compared to the target-side GEDs, the language models are more sensitive to hyper-parameters, particularly to the variability $\beta$. Specifically, on the English dataset, when variability $\beta$ is small, meaning that the $\lambda$ is almost the same at different decoding time steps, the language model not only fails to gain improvement but even leads to a decrease in the $F_{0.5}$ score when $\alpha$ is large. Although the target-side GEDs are robust to the hyper-parameters, they tend to perform better with a larger $\alpha$ and smaller $\beta$ in English, and with a moderate $\alpha$ and larger $\beta$ in Chinese.

## 6 Related Works

### 6.1 Grammatical Error Correction

There exist two main approaches: **sequence-to-sequence** (Seq2Seq) and **sequence-to-edit** (Seq2Edit). The Seq2Seq-based approach regarding GEC as a monolingual machine translation task,

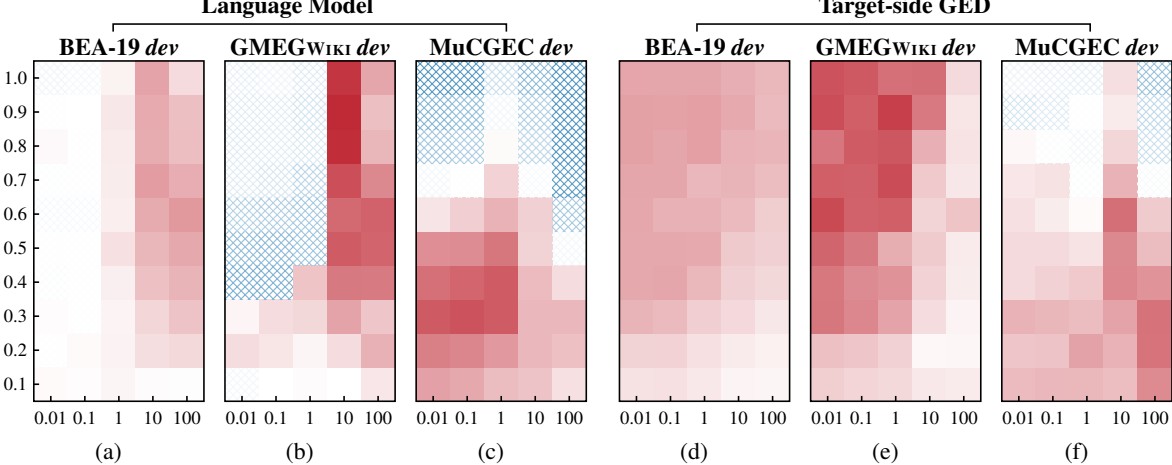

Figure 2: Model performance ($\mathbf{F}_{0.5}$) of decoding intervention compared to vanilla decoding with different scale $\alpha$ and variability $\beta$. The $x$-axis is $\beta$ and $y$-axis is $\alpha$. The red cells ■ denote superior performance of decoding intervention compared to the vanilla decoding. The blue cells ⊠ represent inferior performance. A deeper color indicates a larger performance difference.

is the most widely used approach in the GEC community recently. Though the Seq2Seq-based approach has achieved the state-of-the-art (SOTA) performance on various benchmarks (Sun et al., 2021; Rothe et al., 2021; Zhang et al., 2022b, *inter alia*), they typically have a slow inference speed due to their autoregressive decoding process.

In order to deal with the slow inference speed of the Seq2Seq-based approach, numerous recent works focus on the second approach, the Seq2Edit-based approach (Gu et al., 2019; Awasthi et al., 2019; Omelianchuk et al., 2020; Zhang et al., 2023, *inter alia*). This approach regards GEC as a sequence labeling task. A Seq2Edit model is trained to predict the edit operations (e.g., keep, insert, delete, replace) for each token in the input sentence to transform it into a correct one. The most representative work, GECToR (Omelianchuk et al., 2020), achieves a comparable performance to the state-of-the-art Seq2Seq approach, with a 10x faster inference speed.

## 6.2 Decoding Intervention

The idea of decoding interventions has been widely used in many NLP tasks. Existing works can be categorized into two temporal stages: early and contemporary.

Early-stage works are mainly used to improve the performance of Seq2Seq-based approaches by using a language model trained on a large amount of monolingual data to intervene in the decoding process remedying the lack of parallel data

(Gülçehre et al., 2015; Kannan et al., 2018; Zhao et al., 2019, inter alia). To the best of our knowledge, these early-stage works mainly focus on tasks like machine translation and automatic speech recognition, with no known attempts to apply them to GEC. This kind of decoding intervention has become less popular in recent years, as the advent of powerful pre-trained models has largely mitigated the problem of lacking parallel data.

Recent works, on the other hand, mostly focus on using decoding interventions to steer pre-trained language models towards generating desired outputs, such as certain topics, sentiments, or the avoidance or inclusion of specific words (Dathathri et al., 2020; Krause et al., 2021; Liu et al., 2021; Chen et al., 2022, inter alia).

Our work shares similarities with early-stage works in that we also use a language model to intervene in the decoding process. However, we distinguish ourselves by focusing on the GEC task and proposing the use of a target-side GED model to incorporate explicit grammaticality awareness into the decoding process. It is worth noting that there is a work conducts a decoding intervention in GEC (Sun and Wang, 2022). However, their motivation is to adjust the precision-recall trade-off.

## 7 Conclusions

In this paper, we propose a unified decoding intervention framework for GEC models. Within this framework, we discover and investigate two useful

critics: the language model critic and the target-side GED critic. Among them, the target-side GED critic represents a novel contribution. While most existing research has employed GED on the input side, this work is the first to leverage GED on the target side to assist GEC. Although the concept of a language model critic may not be entirely new, we argue that it is still worth investigating its effectiveness on the GEC task, especially in the era of pre-trained language models.

Experiments conducted on four English and Chinese GEC datasets lead to several promising findings. Firstly, the decoding intervention framework can consistently and substantially improve the performance of GEC models, regardless of whether a language model or error detector is used as the critic. Secondly, The language model critic is better at improving the recall rate, while the target-side GED critic is better at improving the precision rate. Thirdly, while the size of the critic has a minor impact on the ESL dataset, it becomes substantial on the multi-domain English dataset from native speakers, as well as the Chinese dataset. Finally, aided by the decoding intervention framework, our baseline GEC model shows competitive performance when compared to state-of-the-art models.

## Limitations

The use of the critic introduces additional computational costs and GPU memory usage. Consequently, the decoding intervention has slower decoding speeds than the vanilla decoding, especially in the native writing dataset where a larger critic is required for better performance. In the future, we will further explore methods to reduce the computational costs of the decoding intervention framework, for example, distilling a larger critic into a small one, or using a lightweight mechanism to decide when to use the critic.

Besides, this work primarily focuses on the decoding intervention framework for GEC models. It would be interesting to investigate whether the decoding intervention framework can be applied to other Seq2Seq-based approaches in different NLP tasks, such as machine translation and text summarization, or how to design a suitable critic for these tasks. We leave these questions for future work.

## Acknowledgments

We thank the anonymous reviewers for their valuable comments and suggestions. We are very grateful to Yu Zhang and Yue Zhang for their early-stage exploration on the decoding intervention framework. The code of this paper is largely based on the open-source framework SUPAR[5], which is developed and maintained by Yu Zhang.

This work was supported by National Natural Science Foundation of China (Grant No. 62176173), Alibaba Group through Alibaba Innovative Research Program, and a Project Funded by the Priority Academic Program Development (PAPD) of Jiangsu Higher Education Institutions.

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

| Dataset | #Sentences | %Error | Usage |
|---|---|---|---|
| CLang8 | 2,372,119 | 57.8 | Pre-training |
| FCE | 34,490 | 62.6 | Fine-tuning I |
| NUCLE | 57,151 | 38.2 | Fine-tuning I |
| W&I+LOCNESS | 34,308 | 66.3 | Fine-tuning I&II |
| BEA-19 *Dev* | 4,384 | 65.2 | Validation |
| GMEG_WIKI *Dev* | 992 | – | Validation |
| CoNLL-14 *Test* | 1,312 | 72.3 | Testing |
| BEA-19 *Test* | 4,477 | – | Testing |
| GMEG_WIKI *Test* | 992 | 82.3 | Testing |

Table 6: Statistics of English GEC datasets. **#Sentences** denotes the number of sentences. **%Error** refers to the proportion of erroneous sentences.

| Dataset | #Sentences | %Error | Usage |
|---|---|---|---|
| Lang8 | 1,220,906 | 89.5 | Training |
| HSK | 15,6870 | 60.8 | Training |
| MuCGEC *dev* | 1,125 | 95.1 | Validation |
| MuCGEC *test* | 5,938 | 92.2 | Testing |

Table 7: Statistics of Chinese GEC datasets.

## A  Dataset Statistics

The information of all datasets used in our English and Chinese experiments is listed in Table 6.

## B  Details of GEC Model

### B.1  Hyper-parameters

The main hyper-parameters adopted by our GEC are presented in Table 8. We use the `facebook/bart-large` for English and `fnlp/bart-large-chinese` for Chinese. For fine-tuning BART on GEC data, we directly utilize the same hyper-parameters described in Katsumata and Komachi (2020). When confronting sentences longer than the max input length, we keep them unchanged during predicting.

### B.2  Train procedure

We adopt a three-stage finetuning strategy for our English GEC model, following Omelianchuk et al. (2020):

STAGE 1: We train GEC model on the cleaned version of the Lang8 dataset (CLang8) released by Rothe et al. (2021);

STAGE 2: We fine-tune the model on FCE dataset (Yannakoudakis et al., 2011), NUCLE dataset (Dahlmeier et al., 2013) and the W&I + LOCNESS train-set (Bryant et al., 2019);

STAGE 3: We further fine-tune the model on the W&I + LOCNESS test set only.

For *Chinese*, we use Chinese Lang8 dataset (Zhao et al., 2018) and HSK dataset as the finetuning data, following Zhang et al. (2022b).

| Configuration | Value |
|---|---|
| **Fine-tuning** | |
| Pre-trained Model | BART-large (Lewis et al., 2020) |
| Number of epochs | 60 |
| Devices | 2 Tesla V100 GPU (32GB) |
| Batch size | 40960 (E); 16384 (C) tokens |
| Update frequency | 10 |
| Optimizer | AdamW (Loshchilov and Hutter, 2019) ($\beta_1 = 0.9, \beta_2 = 0.999, \epsilon = 1 \times 10^{-8}$) |
| Max input length | 64 (E); 128 (C) |
| Loss function | Label smoothed cross entropy (label-smoothing=0.1) (Szegedy et al., 2016) |
| Dropout | 0.3 |
| Weight decay | $1 \times 10^{-2}$ (E); 0 (C) |
| Gradient clip | 0.1 (E); 1.0 (C) |
| Learning rate (E Stage 1) | $3 \times 10^{-5}$ |
| Learning rate (E Stage 2) | $5 \times 10^{-6}$ |
| Learning rate (E Stage 3) | $3 \times 10^{-6}$ |
| Learning rate (C) | $3 \times 10^{-6}$ |
| Warmup updates (E Stage 1) | 2000 |
| Warmup updates (E Stage 2 & 3) | 0 |
| Warmup updates (C) | 2000 |
| **Generation** | |
| Beam size | 12 |
| Max input length | 64 (E); 128 (C) |

Table 8: Hyper-parameter values used in our GEC model. E and C denote the English and Chinese model, respectively.

### B.3  Tricks

As shown in the Table 2, our baseline model outperforms most previous works on two datasets: CoNLL-14 and MuCGEC. The reasons are as follows:

**CoNLL-14**: We use the same hyperparameters as Katsumata and Komachi (2020), but different from them, we follow the suggestion of Rothe et al. (2021) to post-process the model's output on CoNLL-14 to make its tokenization more consistent with the evaluation data. For instance, we remove spaces in Hyphenated words like "face - to - face" to produce "face-to-face". We implemented this post-processing step after observing that our GED critic performs better than vanilla decoding for this type of tokenization error, thereby isolating improvements not attributable to syntactic factors.

**MuCGEC**: The improvement of our baseline model on this dataset comes from two aspects: While we employ the same "`fnlp/bart-large-chinese`" as previous works, the model's parameters were updated earlier this year, resulting in a substantial performance boost (from 46.51 to 47.04). We noticed that the Chinese correction model sometimes produces repeated tokens during decoding, e.g., Input: "为

怎么呢？", Output: "为什么呢？为什么呢？".
To counter this, we use a simple decoding rule
that limits the model's output to no more than 1.8
times the length of the input, leading to further
improvement (from 47.04 to 47.41). A better
solution, as suggested by Jiang et al. (2023), is to
filter out the instances of which the output length
is more than 1.5 times the input length during
training.

## C   Details of Target-side GED Critic

### C.1   Hyper-parameters

During training, we use the AdamW optimizer
(Loshchilov and Hutter, 2019) with a weight de-
cay of 0.01, $\beta_1 = 0.9$ and $\beta_2 = 0.98$. The BART
model and the classifier have a learning rate of
$5 \times 10^6$ and $2.5 \times 10^7$, respectively. We use an expo-
nential decay scheduler with a decay rate of 0.75
and a decay step of 5000. The batch size is 65536
tokens, and the maximum sequence length is 1024.
The dropout rate is set to 0.3, and the label smooth-
ing rate is set to 0.01. The maximum number of
epochs is set to 5.