# OpenReview forum: "Improving Seq2Seq Grammatical Error Correction via Decoding Interventions"
_EMNLP/2023/Conference — EMNLP 2023 Findings_

### Official Review · Reviewer_wsFU · 2023-08-01

**Soundness:** 3

**Excitement:**

2: Mediocre: This paper makes marginal contributions (vs non-contemporaneous work), so I would rather not see it in the conference.

**Paper Topic And Main Contributions:**

This paper proposes a decoding intervention framework for improving the sequence-to-sequence (Seq2Seq) approach to grammatical error correction (GEC). The main idea is to use an external critic model to dynamically evaluate and influence the next token generated by the GEC model during decoding. The paper investigates two types of critics: a pre-trained left-to-right language model and an incremental target-side grammatical error detector. The paper conducts experiments on four English and Chinese GEC datasets and shows that the decoding intervention framework can consistently and substantially improve the performance of GEC models, achieving results competitive with state-of-the-art methods.

**Reasons To Accept:**

The GED critic sounds interesting and may potentially help improve the performance.

**Reasons To Reject:**

1. LM-critic is not new. It was proposed as early as 2016~2018 which adopts a n-gram LM or a neural LM to improve the GEC performance. See [1], [2].

2. I don’t think using a GED model trained on the same data as a critic for GEC would introduce more information, because the information they have is the same or similar. Of course, due to the size limitations of the GEC model, it has to remember more things, so it may overlook some details at the grammar level, and these details can be well grasped by the GED model. But if the model size is further increased, I believe that the GEC model itself can do this well enough. We can see that the most advanced generative models (GPT-3.5, GPT-4) are all decoder-only unidirectional attention layers, but they do not have grammar-level problems, which also fully illustrates this point. What I want to say is that I do not deny that using an additional GED to assist decoding may improve performance when the model is small, but the author must also be clear that this will bring additional time overhead. If we allow the same additional time overhead and make the GEC model larger, what will be the effect? I think it may not be worse than adding a GED - at least the author should add an experiment to justify the practical value of this method.

[1] https://aclanthology.org/N18-2046/
[2] https://aclanthology.org/P18-1097/

**Reproducibility:**

3: Could reproduce the results with some difficulty. The settings of parameters are underspecified or subjectively determined; the training/evaluation data are not widely available.

**Reviewer Confidence:**

5: Positive that my evaluation is correct. I read the paper very carefully and I am very familiar with related work.

---

> ### Author Rebuttal · Authors · 2023-08-29
>
> 1. **LM-critic is not new. It was proposed as early as 2016~2018 which adopts a n-gram LM or a neural LM to improve the GEC performance. See [1], [2].**
>
>     The main contribution of our work is that we propose a unified decoding intervention framework.
>     Within this framework, we discover and investigate two useful critics: the language model critic and the target-side GED critic.
>
>     Although the concept of a language model critic may not be entirely new, we argue that it is still worth investigating its effectiveness on the GEC task, especially in the era of pre-trained language models.
>     In addition, the language model critic and the target-side GED critic have complementary effects, and combining them can achieve greater improvement.
>
>     Notably, our utilization of the language model critic diverges from previous approaches.
>     Our early trials show that existing methods not only failed to improve performance but sometimes lead to inferior performance.
>     From Table 5 and Figure 2, we can see that the dynamic coefficient $\beta$ is the key to the success of our method.
>
> 2. **I don’t think using a GED model trained on the same data as a critic for GEC would introduce more information, because the information they have is the same or similar.**
>
>     We would like to clarify this.
>
>     Please kindly note that the training data used for the target-side GED model is different from that used for GEC model.
>     Specifically, our target-side GED model requires the presence of errors in the target sentences.
>
>     Consider the following input-output pair for the target-side GED model: Input: `{source="The quick brown fox jump over the lazy dog", generated_tokens="The quick brown fox", next_token="runs"}`, Output label: `"substitution error"`. It is essential for an error to be present in the `"next_token"` field, as otherwise, the GED model would invariably predict a `"correct"` label.
>
>     Given that target sentences in GEC training data are grammatically correct, they are unsuitable for training our target-side GED model.
>
>     To address this issue, we use the baseline GEC model to generate 12 output sentences which may be erroneous via beam search.
>     Then, we label the errors in each token by employing an edit distance algorithm.
>
> 3. **If we allow the same additional time overhead and make the GEC model larger, what will be the effect? I think it may not be worse than adding a GED - at least the author should add an experiment to justify the practical value of this method.**
>
>     First, it is worth noting that our preliminary experiments indicate that a larger GED model does not necessarily yield better results.
>     For example, in CoNLL-14 dataset, our model (**400M+400M**) outperforms the much larger T5-xxl model with **11B** parameters, as proposed by Rothe et al. (2021).
>
>     Secondly, the cost-effectiveness of scaling up the GED model's parameters appears to be limited. Rothe et al. (2021) reported a parameter increase from **770M** to at least **3B**, merely to achieve an $F_{0.5}$ score improvement from $66.10$ to $67.75$ on CoNLL-14 and from $72.06$ to $73.92$ on the BEA test.
>     In contrast, we argue that the addition of a smaller **400M** GED model during decoding incurs a much more reasonable cost.
>
>     Finally, our decoding intervention is not in conflict with deploying a larger GED model.
>     We can also add a GED-critic to a larger GEC model to further improve its performance.
>
>     However, it should be noted that due to computational constraints, we were unable to conduct experiments with a larger GEC model.
>
> ***
>
> [A Simple Recipe for Multilingual Grammatical Error Correction](https://aclanthology.org/2021.acl-short.89) (Rothe et al., ACL-IJCNLP 2021)

---

### Official Review · Reviewer_cq4q · 2023-08-05

**Soundness:** 3

**Excitement:**

3: Ambivalent: It has merits (e.g., it reports state-of-the-art results, the idea is nice), but there are key weaknesses (e.g., it describes incremental work), and it can significantly benefit from another round of revision. However, I won't object to accepting it if my co-reviewers champion it.

**Paper Topic And Main Contributions:**

The paper introduces a decoding intervention framework that utilizes an external critic to assess the suitability of tokens generated during decoding. Two types of critics are investigated: a pre-training language model critic and a grammatical error detector critic.

Strengths:

The proposed GEC model demonstrates comparable performance to state-of-the-art models across all datasets.
The paper conducts numerous ablative experiments to explore the effects of different model sizes and coefficients on the results.
Weaknesses:

The motivation behind the design of the lambda formula remains unclear and needs further explanation.
On line 294, it is suggested that Beta should be a static coefficient rather than a dynamic one to enhance clarity and consistency.
The method proposed in the paper lacks notable innovation, as similar works have been previously published in the field. Further emphasis on the novel aspects of the approach could strengthen the paper's contribution.

**Questions For The Authors:**

1. Why does the baseline performance in the paper appear to be significantly higher than that in most prior related works?

**Reasons To Accept:**

1. The proposed GEC model exhibits commendable performance, achieving results on par with state-of-the-art models across all datasets.

2. The paper demonstrates a rigorous approach by conducting numerous ablative experiments, effectively exploring the impacts of various model sizes and coefficients on the results. This thorough investigation adds depth to the study and enhances the credibility of the findings.

**Reasons To Reject:**

1. The paper lacks clarity in explaining the motivation behind the design of the lambda formula, which should be further elaborated to provide a better understanding of its purpose and significance.

2. On line 294, it is recommended that Beta should be treated as a static coefficient instead of a dynamic one to enhance the consistency and comprehensibility of the approach.

3. While the paper presents a method, it fails to showcase notable innovation, as several similar works have been previously published. The paper should emphasize and highlight distinct features or improvements over existing approaches to strengthen its contribution.

**Reproducibility:**

3: Could reproduce the results with some difficulty. The settings of parameters are underspecified or subjectively determined; the training/evaluation data are not widely available.

**Reviewer Confidence:**

3: Pretty sure, but there's a chance I missed something. Although I have a good feel for this area in general, I did not carefully check the paper's details, e.g., the math, experimental design, or novelty.

---

> ### Author Rebuttal · Authors · 2023-08-29
>
> 1. **The paper lacks clarity in explaining the motivation behind the design of the lambda formula, which should be further elaborated to provide a better understanding of its purpose and significance.**
>
>     The motivation behind the design of the lambda formula is to mitigate the impact of potential errors made by the critic.
>
>     Given that critics are not infallible, we require a mechanism to assess the reliability of a critic's predictions and to adjust its influence dynamically.
>     Previous research on model calibration has shown that high confidence in a model's prediction is often indicative of accuracy. This strong correlation between a model's confidence and the accuracy of its predictions led us to use the entropy of the critic's prediction as a proxy for estimating its confidence.
>     Rather than relying on the absolute value of entropy, we employ the entropy ratio between the critic and the GEC model. This approach allows the critic to exert greater influence when its confidence exceeds that of the GEC model and lesser influence otherwise.
>     Finally, to ensure that the critic's influence remains within an acceptable range, we introduce the $\beta$ coefficient.
>
> 2. **On line *294*, it is recommended that Beta should be treated as a static coefficient instead of a dynamic one to enhance the consistency and comprehensibility of the approach.**
>
>     Thank you for your suggestion, we will improve the paper for this issue.
>
> 3. **While the paper presents a method, it fails to showcase notable innovation, as several similar works have been previously published. The paper should emphasize and highlight distinct features or improvements over existing approaches to strengthen its contribution.**
>
>     The main contribution of our work is that we propose a unified decoding intervention framework.
>     Within this framework, we discover and investigate two useful critics: the language model critic and the target-side GED critic.
>
>     Among them, the target-side GED critic represents a novel contribution.
>     While most existing research has employed GED on the input side, this work is the first to leverage GED on the target side to assist GEC.
>
>     Although the concept of a language model critic may not be entirely new, we argue that it is still worth investigating its effectiveness on the GEC task, especially in the era of pre-trained language models.
>
>     In addition, the language model critic and the target-side GED critic have complementary effects, and combining them can achieve greater improvement.
>
>     We will clarify these contributions in the next version of the paper.
>
> 4. **Why does the baseline performance in the paper appear to be significantly higher than that in most prior related works?**
>
>     As shown in the Table 2, our baseline model outperforms most previous works across multiple datasets: CoNLL-14, GMEG-wiki, and MuCGEC.
>
>     The reasons are as follows:
>
>     - CoNLL-14: We use the same hyperparameters as Zhang et al. (2022), but different from them, we follow the suggestion of Rothe et al. (2021) to post-process the model’s output on CoNLL-14 to make its tokenization more consistent with the evaluation data. For instance, we remove spaces in Hyphenated words like "face - to - face" to produce "face-to-face".
>         We implemented this post-processing step after observing that our GED critic performs better than vanilla decoding for this type of tokenization error, thereby isolating improvements not attributable to syntactic factors.
>     - GMEG-wiki: The only previous result on this dataset is from Yasunaga et al. (2021), who trained their model on synthetic data only.
>         This makes direct comparisons less meaningful.
>         We will include a clear note regarding this in the next version of the paper.
>     - MuCGEC: The improvement on this dataset comes from two aspects:
>          1) While we employ the same ``fnlp/bart-large-chinese'' as previous works, the model's parameters were updated earlier this year, resulting in a substantial performance boost (from 46.51 to 47.04).
>          2) We noticed that the Chinese correction model sometimes produces repeated tokens during decoding (e.g., Input: "为怎么呢？", Output: "为什么呢？为什么呢？").
>         To counter this, we use a simple decoding rule that limits the model's output to no more than 1.8 times the length of the input, leading to further improvement (from 47.04 to 47.41).
>
>
> ***
>
> - [SynGEC: Syntax-Enhanced Grammatical Error Correction with a Tailored GEC-Oriented Parser](https://aclanthology.org/2022.emnlp-main.162) (Zhang et al., EMNLP 2022)
> - [A Simple Recipe for Multilingual Grammatical Error Correction](https://aclanthology.org/2021.acl-short.89) (Rothe et al., ACL-IJCNLP 2021)
> - [LM-Critic: Language Models for Unsupervised Grammatical Error Correction](https://aclanthology.org/2021.emnlp-main.611) (Yasunaga et al., EMNLP 2021)

---

### Official Review · Reviewer_izec · 2023-08-14

**Soundness:** 4

**Excitement:**

4: Strong: This paper deepens the understanding of some phenomenon or lowers the barriers to an existing research direction.

**Justification For Ethical Concerns:**

no ethical concerns are apparent

**Paper Topic And Main Contributions:**

This paper is situated in the research are of Grammar Error Correction (GEC). Specifically it proposes modifications for a seq2seq neural model, in order to improve GEC results.

**Questions For The Authors:**

The paper very much brags about  'state-of-the-art (SOTA)'.
However, looking at best results, it would be interesting to have a note (just a note) whether this level of SOTA can be useful for practical applications.

The paper mentions 'they typically have a slow inference speed' (line 496).
What about the presented model - is its speed slow?
Is such a concern important for research systems?

**Reasons To Accept:**

The idea of this paper is to consider two interventions in the decoder stage of the model. One intervention uses a large left-to-right language model for predicting the next word. The other intervention is to integrate a Grammar Error Detection (GED) module, which just labels whether the next-word-to-be-produced by GEC would be correct relative to the whole sentence.
This proposal is interesting and worthy of investigation.

The paper describes the implementation of those ideas and their extensive testing with several published datasets, including in English and Chinese. Ablation studies and qualitative analysis are also included.

Overall the results  indicate that the proposed modifications are useful, as they manage to slightly improve the results relative to previous work with the same datasets.

The paper is well-written and the exposition is very clear.

**Reasons To Reject:**

N/A

**Reproducibility:**

4: Could mostly reproduce the results, but there may be some variation because of sample variance or minor variations in their interpretation of the protocol or method.

**Reviewer Confidence:**

3: Pretty sure, but there's a chance I missed something. Although I have a good feel for this area in general, I did not carefully check the paper's details, e.g., the math, experimental design, or novelty.

---

> ### Author Rebuttal · Authors · 2023-08-29
>
> 1. **The paper very much brags about 'state-of-the-art (SOTA)'. However, looking at best results, it would be interesting to have a note (just a note) whether this level of SOTA can be useful for practical applications.**
>
>     Thank you for your suggestion, we will add relevant content in the next version.
>
> 2. **The paper mentions 'they typically have a slow inference speed' (line 496). What about the presented model - is its speed slow? Is such a concern important for research systems?**
>
>     In this paper, we still use the Seq2Seq model, which is auto-regressive during  the prediction phase, so the inference speed is relatively slow.
>     But for research experiments, this speed is acceptable.
>     However, some papers may focus on how to speed up inference for practical application, such as online editor.

---

### Official Review · Reviewer_hcJu · 2023-08-16

**Soundness:** 4

**Excitement:**

2: Mediocre: This paper makes marginal contributions (vs non-contemporaneous work), so I would rather not see it in the conference.

**Paper Topic And Main Contributions:**

The paper proposes to use an external pre-trained LM as a critic to intervene in the seq2seq decoding process. The critic dynamically influences the choice of the next token by producing a penalty score to the probability distribution of the GEC model. The authors investigate two kinds of pre-trained LM, GPT2 and BART, and conduct experiments on both English and Chinese datasets to verify the efficacy.

**Questions For The Authors:**

1. Does the subword tokenizer need to be the same for the GEC model and critic models? If the vocabulary is different, how can they be added together in the probability space?
2. I am curious about the two-stage GEC experiment results, which means performing the target-side GEC directly on the vanilla decoding outputs.

**Reasons To Accept:**

1. The paper proposes to use an external pre-trained LM to further correct the generated sentence in the decoding process dynamically, which is a potential method to scale up to LLM and get further improvement.

**Reasons To Reject:**

1. Not robust enough. The dynamic coefficient sometimes may lead to a decline in the Precise and Recall.
2. The improvement from gpt2-large to gpt2-xl is not significant, which makes me doubt the potential improvement brought by stronger LM.
3. The critic needs to be finetuned on the specific GEC dataset.

**Reproducibility:**

3: Could reproduce the results with some difficulty. The settings of parameters are underspecified or subjectively determined; the training/evaluation data are not widely available.

**Reviewer Confidence:**

4: Quite sure. I tried to check the important points carefully. It's unlikely, though conceivable, that I missed something that should affect my ratings.

---

> ### Author Rebuttal · Authors · 2023-08-29
>
> 1. **Not robust enough. The dynamic coefficient sometimes may lead to a decline in the Precise and Recall.**
>
>     We are unclear as to which experimental results you are referring to.
>
>     In fact, the dynamic coefficient serves as a balancing mechanism and, as demonstrated in Table 5, and does not lead to a simultaneous reduction in both precision and recall.
>     As mentioned in Section 4.1, the language model critic is better at improving the recall rate, whereas the target-side GED critic is better at improving the precision rate.
>     However, improving one score often leads to a decline in the other.
>     The role of the dynamic coefficient is to balance these opposing influences, enabling the model to achieve a better $F_{0.5}$ score.
>
> 2. **The improvement from gpt2-large to gpt2-xl is not significant, which makes me doubt the potential improvement brought by stronger LM.**
>
>     The observation that the improvement from gpt2-large to gpt2-xl is not significant can be explained by our decoding intervention framework.
>
>     The main prediction is made by the GEC model, which contains only 400 million parameters.
>     For context, this is $0.52$x and $0.27$x the size of gpt2-large and gpt2-xl, respectively.
>     The language model is only an intervention critic, designed to further improve the performance  of an existing GEC model without requiring any **re-training** or **fine-tuning**.
>
>     Therefore, the extent to which the language model critic can improve performance is limited by the capabilities of the GEC model itself.
>
> 3. **The critic needs to be finetuned on the specific GEC dataset.**
>
>     In this paper, we discover and investigate two useful critics.
>
>     The *first* is the target-side GED critic, which does require training, albeit on a GED dataset rather than a GEC dataset.
>     We believe this is acceptable, particularly given the absence of an off-the-shelf target-side GED model.
>
>     The *second* critic is the language model critic, which is a left-to-right pre-trained language model like GPT2.
>     For this critic, there is **no** need for, **nor** have we performed, any fine-tuning on the GEC dataset.
>
> 4. **Does the subword tokenizer need to be the same for the GEC model and critic models? If the vocabulary is different, how can they be added together in the probability space?**
>
>     In this paper, we use the same subword tokenizer for the GEC model and the critic models for the sake of convenience.
>
>     However, it's worth noting that using the same subword tokenizer is not a strict requirement.
>     Some simple **engineering solutions** can address the situation where the subword tokenizer is different.
>     For instance, we can change the penalty from a subword level to a word level.
>
> 5. **I am curious about the two-stage GEC experiment results, which means performing the target-side GEC directly on the vanilla decoding outputs.**
>
>     What do you mean by target-side GEC? Our GEC model performs correction to the source-side input sentence at a single stage.

---

### Meta-Review · Area_Chair_g8XU · 2023-09-19

**Recommendation:** 3

**Metareview:**

This paper proposes a decoding intervention framework for Grammatical Error Correction (GEC), where they employ an external critic to improve the choice of the next token. They investigate 2 types of critics – pre-trained LMs and a target-side Grammatical Error Detection (GED) critic that requires training. Experimental results on English and Chinese benchmarks are presented that show improvements over the baseline approaches and demonstrate performance competitive with SOTA models.

The paper is thorough and the experiments demonstrate the effectiveness of the approach.

While the study is interesting, and the experiments are thorough, the main issue pointed out in the reviews is the lack of novelty. In particular,  one of the two critic components -- using a language model to improve GEC performance -- has been studied before.

The reviewers also express concern regarding the utility of the approach in practical applications, given that adding a critic increases the model complexity and thus is expected to increase decoding time. It would also be useful to see, given these complexity concerns,  whether increasing the GEC model size would have a similar effect, as adding the critic component.

---

### Decision · Program_Chairs · 2023-10-07

**Decision:**

Accept-Findings

**Comment:**

This paper proposes a decoding intervention framework for Grammatical Error Correction (GEC), where they employ an external critic to improve the choice of the next token. They investigate 2 types of critics – pre-trained LMs and a target-side Grammatical Error Detection (GED) critic that requires training. Experimental results on English and Chinese benchmarks are presented that show improvements over the baseline approaches and demonstrate performance competitive with SOTA models.

The paper is thorough and the experiments demonstrate the effectiveness of the approach.

While the study is interesting, and the experiments are thorough, the main issue pointed out in the reviews is the lack of novelty. In particular,  one of the two critic components -- using a language model to improve GEC performance -- has been studied before.

The reviewers also express concern regarding the utility of the approach in practical applications, given that adding a critic increases the model complexity and thus is expected to increase decoding time. It would also be useful to see, given these complexity concerns,  whether increasing the GEC model size would have a similar effect, as adding the critic component.